# Farmers' Willingness to Accept Compensation to Maintain the Benefits of Urban Forests

**Xueyan Wang [1], Jan F Adamowski [2], Guangda Wang [3], Jianjun Cao [1,*], Guofeng Zhu [1], Junju Zhou [1], Chunfang Liu [4,5] and Xiaogang Dong [1]**

[1] College of Geography and Environmental Science, Northwest Normal University, Lanzhou 730070, China
[2] Department of Bioresource Engineering, Faculty of Agricultural and Environmental Sciences, McGill University, Sainte Anne de Bellevue, QC H9X 3V9, Canada
[3] Department of Mathematical Sciences, Xi'an Jiaotong-Liverpool University, Suzhou 215123, China
[4] College of Social Development and Public Administration, Northwest Normal University, Lanzhou 730070, China
[5] Gansu Engineering Research Center of Land Utilization and Comprehension Consolidation, Lanzhou 730070, China
* Correspondence: caojj@nwnu.edu.cn; Tel.: +86-0931-7971565

**Abstract:** The Returning Farmland to Forest Program (RFFP) was implemented in China in 1999 with the goal of supporting environmental restoration by returning significant areas of cultivated land to forest. While afforestation supports long-term ecosystem services like carbon sequestration and the reduction of soil and water loss, it also reduces the amount of available arable land, putting financial pressure on those who depend on it for their livelihoods. In an effort to balance both ecological and economic benefits, regional governments offer financial compensation to farmers to offset these pressures in the form of a dollar amount per hectare of reforested land. The current study explores participants' willingness to accept pay (WTA), along with the difference between the offered per hectare compensation and the amount deemed acceptable by RFFP participants in the region. To this end, 92 households from the representative afforestation area were surveyed in Huining County, Gansu Province, China - an area of strategic ecological importance in the Loess Plateau. The results showed 12.0% of the surveyed respondents to be satisfied with the current compensation policy, while 88.0% of respondents were not. The respondents' lower and upper WTA limits were $221/ha/year and $1331/ha/year, respectively, with an average WTA of $777/ha/year. The compensation that respondents would be most willing to accept was distributed in the $444–888/ha/year and the $889–1331/ha/year ranges, accounting for 37.0% and 31.5% of the total responses, respectively. Gender, age, and education were found to be the main factors influencing a respondents' WTA. Results of the survey suggest that the actual compensation amount ($355/ha/year) is much lower than respondents' WTA, and that compensation measures and policies should be improved to guarantee a basic income.

**Keywords:** willingness to accept pay; afforestation; rural development; ecological benefits; economic benefits

## 1. Introduction

It is well known that climate change contributes to water and food scarcity, biodiversity loss, and the uncertainty of future generations. To minimize these threats, several mitigation strategies have been introduced, including enhancing afforestation and improving forest management practices [1–3]. In the 1920s, the USA became the first country to propose a policy to facilitate afforestation [4]. France, Britain, Germany, and other European countries soon followed, beginning to return farmland to forest and grassland in the 1960s [5]. Since the 1998 Yangtze River flood, environmental protection

has become a significant concern for the Chinese government. The Returning Farmland to Forest Program (RFFP) was implemented in 1999 with the goal of returning $3.2 \times 10^7$ ha of arable land to forestland by 2010 [6,7]. Land with a slope greater than 25° was targeted, as conversion of this land was considered to have maximum potential for controlling soil erosion, alleviating poverty, and transforming rural livelihoods. From 1998 to 2008, China saw a $4.2 \times 10^7$ ha increase in forest area, contributing significantly to the worldwide forested area increase from $1.2 \times 10^8$ ha in 1981 to approximately $2.1 \times 10^8$ ha by the end of 2014 [8]. With forested area increasing, many ecological services such as soil and water conservation, biodiversity maintenance, and climate change mitigation have improved [9–11]. However, one of the main challenges of maintaining afforestation benefits is providing appropriate compensation to farmers [12], especially given that, in contrast to most other forestry projects, the main participants in the RFFP are farmers [13,14], and arable land remains a primary source of income for many rural households, including migrants that lose their jobs in urban areas in China [15,16]. Appropriate compensation can help to balance the ecological and economic benefits of farmland afforestation projects, which have been shown to improve participant awareness and support for environmental protection, and to promote coordination and stable development of society and the economy [17]. In the case of the RFFP, farmer compensation has received a great deal of attention [14,18,19]. Many studies [20,21] have argued that the government compensation standard is arbitrary and insufficient, and that it should be revised to account for regional variations.

At present, two methods are primarily used to estimate appropriate compensation for farmers: Payment for Ecosystem Services (PES) and the Contingent Valuation Method (CVM). The PES method rewards resource managers for the provision of ecosystem services through cash payments or other compensation based on the market, and is characterized by ecological functions subject to trade, the standard of exchange, and supply and demand between the seller and buyer of these services [22–24]. Although it has been widely used to estimate compensation [23], Wegner and Irene [25] found two major obstacles in the design and implementation of effective PES schemes. The first is that PES fails to account for both the physical properties of ecosystem services and the factors influencing the capacity of land users and ecosystem service beneficiaries to participate in PES and derive benefits from it. According to the researchers, this arises from the focus of PES on maximization of economic efficiency, and the assumption that individuals respond to financial incentives uniquely on the basis of personal utility calculations. The second obstacle is that PES schemes tend to be disjointed from broader national strategies for rural development and from wider socio-economic trends that strongly influence the capacity of land users to participate in these schemes and shift to sustainable land-use practices. In practice, both obstacles are difficult to overcome. In contrast to PES, CVM is a survey-based valuation approach allowing for a precise focus on respondents' answers [26–28], and can be leveraged to provide important relevant information for decision makers by presenting respondents a contingent situation and information regarding the willingness to secure changes, such as changes in the availability of public goods, amenities, the quality of commodities, or to adopt changes, such as land use change [14,29–31]. CVM has been extensively reviewed [32,33] as an applied method for assessing the worth of non-market goods and services [34]. CVM can also be used to assess market-based goods, such as livestock, if reducing reliance on these goods can enhance public ecosystem services [35]. As a market simulation method, CVM has inherent shortcomings [32]; however, it contains mechanisms to reduce the impact of most deviations, such as providing the respondents with detailed information, increasing the credibility and certainty of the content of the questions, and controlling the investigation time of each sample at 20 to 30 min [36–38].

Willingness to pay (WTP) and willingness to accept pay (WTA) can be used, in both approaches, as proxy economic measures of environmental services, with the former being more relevant for buyers of ecosystem services (i.e., 'acquisition'), and the latter being more relevant for sellers of ecosystem services (i.e., 'loss') [14,39–41]. Generally, WTP is lower than WTA [42]. Low WTP is primarily attributed to the limited economic value of ecosystem services provided by individual land owners [23], while higher WTA is due to high costs associated with significant changes in the land

use, and participant identities and lifestyles [43,44]. Based on the perspective of welfare loss, however, in developing countries and underdeveloped regions, such as Huining, labeled as a national-level poor county, it is recommended that the WTA of the affected party be used as a reference for the compensation standard, rather than the WTP [45].

Due to obstacles in PES schemes and the inherent mechanisms in CVM to account for shortcomings, in the current study CVM was selected to (1) explore the WTA of farmers from Huining County participating in the afforestation program; (2) determine the amount of compensation needed to fill the gap between previously offered subsidies and the WTA values determined by the locals; and (3) determine the main factors influencing farmers' WTA.

## 2. Materials and Methods

### 2.1. Study Area

Located in the eastern Gansu Province, China (104°29′–105°31′E, 35°24′–36°26′N) at a mean altitude of 2025 m, Huining County covers an area of roughly 6439 km$^2$, and occupies a strategic position in the ecology of the Loess Plateau [46] (Figure 1). The area experiences mean annual temperatures of 6–9 °C and average annual rainfalls ranging from 180–450 mm. In this area, arable land is the traditional source of income for locals. Encouraged by the national RFFP, afforestation in the region began in 1999, with most trees planted between 2000 and 2003. Since RFFP implementation, arable land in the area has shrunk considerably. By the end of 2015, the county's total afforested area had reached 706.7 km$^2$, representing a forest coverage rate of 12.47%. Due to the quality of seedlings, high sapling maintenance costs and water deficiencies, the rate of tree survival was low in the county's northern and central regions. However, in the southern region of Huining County, which is the study area of the present research, tree survival rate was relatively high due to greater annual precipitation [47].

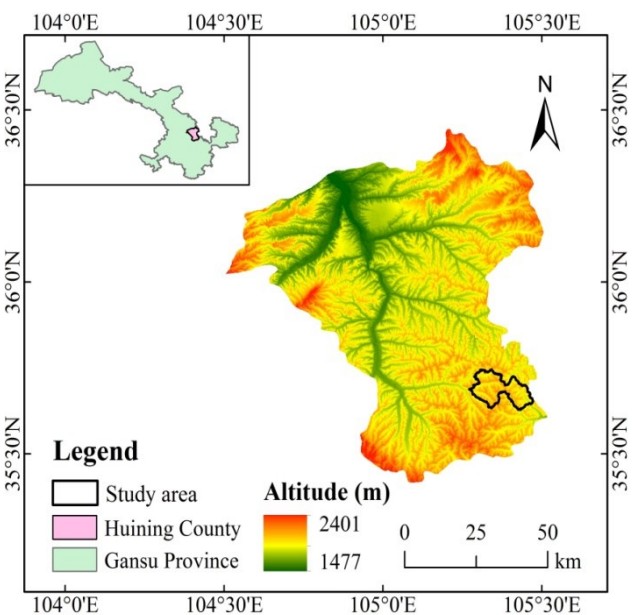

**Figure 1.** The study area in Huining County, Gansu Province, China.

### 2.2. Questionnaire Design and Field Survey

From late July to early August of 2017, a total of 92 households were surveyed. First, 20 households participating in the RFFP and a few local staff members engaged in this project were interviewed with an open-ended questionnaire. Based on these collected data, a semi-structured questionnaire comprising both open-ended and multiple-choice questions was designed to counteract the biases

inherent in single data sources [48]. Next, a pre-survey of 15 households was conducted to identify potential improvements in the developed questionnaire. Finally, after modification and finalization of the questionnaire, the formal investigation was carried out. Roughly 900 households (with an average of five people/household) were located in the study area (Figure 1). In accordance with social survey requirements [35], 100 households were randomly selected as respondents from these 900, accounting for about 10% of the total households. After removing households that had moved or that did not wish to participate, 92 valid surveys remained.

Survey questions fell into several main categories: participant gender, age, educational background, area of total arable land, area of total afforested area, type of developed forest, duration of afforestation efforts, income derived from forests, and government compensation received. Additional questions probed the participants' willingness to accept compensation and to return farmland to forests, as well as the participant awareness of potential environmental problems. Face-to-face interviews were conducted at each stage to ensure a high response rate, maintain respondent motivation, and support the use of graphical supplements [49,50].

*2.3. Data Processing Method*

Several methods have been used to calculate WTA, for example, the individual utility function theory [14], median [51], and Maximum Likelihood Estimation [52]. In the current study, the WTA is expressed as a weighted-average—a simple but accurate approach [53,54]. One-way variance analysis (one-way ANOVA) was used to test the differences between socio-economic variables and WTA values, and the least significant difference test was conducted when significant differences were detected through ANOVA. Multivariate linear regression was used to analyze the effects of socio-economic variables on respondent WTA [18,55], as this approach has been determined to be suitable for analyzing the correlation between dependent and independent variables in similar contexts [56]. A $p < 0.05$ was used as the critical significance threshold to determine differences in the variables. The data were analyzed using SPSS 22.0 (SPSS Inc., Chicago, IL, USA) statistical software.

## 3. Results

*3.1. Socioeconomic Characteristics of Participants*

Table 1 shows the socioeconomic characteristics of the respondents. Among the respondents, 59.8% were male, which exceeded the proportion of females (40.2%). The majority of the respondents were farmers (83.7%), while other occupations accounted for 16.1%. The highest level of education obtained by respondents was primary school at 39.1%, junior high school at 20.7%, senior high school at 5.4%, and college at 4.3%. Illiteracy was prevalent in 30.4% of those surveyed. Overall, 72.8% of the total respondents were between 40 and 70 years old, with 25.0% of the total respondents between 60 and 70 years old. In investigated households, 55.0% of families comprised two to five members, while 42.0% contained five to eight members, and 3.0% encompassed more than eight members.

**Table 1.** Socioeconomic characteristics of the respondents.

| Statistical Indicators | Male (55) | Number of Participants | Proportion (%) | Female (37) | Number of Participants | Proportion (%) |
|---|---|---|---|---|---|---|
| | Under 30 | 2 | 2.2 | Under 30 | 1 | 1.1 |
| | 30–40 | 2 | 2.2 | 30–40 | 2 | 2.2 |
| Age | 40–50 | 7 | 7.6 | 40–50 | 11 | 12.0 |
| | 50–60 | 26 | 28.3 | 50–60 | 10 | 10.9 |
| | 60–70 | 13 | 14.1 | 60–70 | 10 | 10.9 |
| | Above 70 | 5 | 5.4 | Above 70 | 3 | 3.3 |

**Table 1.** *Cont.*

| Statistical Indicators | Male (55) | Number of Participants | Proportion (%) | Female (37) | Number of Participants | Proportion (%) |
|---|---|---|---|---|---|---|
| Educational Background | Illiterate | 7 | 7.6 | Illiterate | 21 | 22.8 |
| | Primary School | 26 | 28.3 | Primary School | 10 | 10.9 |
| | Junior High School | 13 | 14.1 | Junior High School | 6 | 6.5 |
| | High School | 7 | 7.6 | High School | 0 | 0 |
| | Undergraduate | 1 | 1.1 | Undergraduate | 1 | 1.1 |
| Profession | Farmer | 42 | 45.7 | Farmer | 34 | 37.0 |
| | Migrant Worker | 6 | 6.5 | Migrant Worker | 1 | 1.1 |
| | Businessperson | 5 | 5.4 | Businessperson | 1 | 1.1 |

### 3.2. Household Land Resources and Perception of Afforestation

On average, 0.3 ha of farmland land per household, or 0.1 ha per person, was returned to forest under the RFFP. In terms of arable land, 1.2 ha per household or 0.2 ha per person was returned to forest cover. At the time of the current study, the existing cultivated land area was 0.8 ha per household, or 0.2 ha/person. Among the investigated households, 41 households, or 44.6%, retained less than 1.0 ha of arable land, while 38 households, or 41.3%, had between 1.0 and 1.5 ha, and 14.1% of the households owned over 1.5 ha of cultivated croplands (Table 2).

**Table 2.** Respondent land resource information and environmental attitude.

| Statistical Indicators | Classification Indicator | Number of Respondents | Proportion (%) |
|---|---|---|---|
| Arable land (ha) | 0–1 | 41 | 44.6 |
| | 1–1.5 | 38 | 41.3 |
| | >1.5 | 13 | 14.1 |
| Arable land lost during afforestation (ha) | 0–0.3 | 63 | 68.5 |
| | 0.3–0.6 | 16 | 17.4 |
| | >0.6 | 13 | 14.1 |
| Forest species | Apricot (*Prunus armeniaca* L.) | 86 | 93.5 |
| | Poplar (*Populus* L.) | 6 | 6.5 |
| Satisfied | Yes | 11 | 12.0 |
| | No | 81 | 88.0 |
| Income from forest | Yes | 11 | 12.0 |
| | No | 81 | 88.0 |
| Attitude towards environmental change | Positive | 40 | 43.5 |
| | Negative | 39 | 42.4 |
| | No change | 13 | 14.1 |
| Attitude towards species increase | Positive | 0 | 0 |
| | Negative | 92 | 100 |

Approximately 93.5% of the respondents emphasized planting apricot trees when returning farmland to forests due to their economic potential. Almost 100% of the respondents believed that species changes during afforestation could have adverse effects. In addition, 56.5% of the respondents believed the impact of forests on local environment to be negative or negligible (Table 2).

### 3.3. Factors Affecting Participant WTA

Respondents returned a wide range of compensation values, from $0 to $1331/ha/year. Based on these values and their frequencies, the average WTA was calculated at $777/ha/year. Participants were divided into four groups in terms of WTA: $0–221/ha/year, $222–444/ha/year, $445–888/ha/year, and $889–1331/ha/year. It was found that the majority of surveyed respondents reported a WTA

value in the range of \$444–888/ha/year or \$889–1331/ha/year - approximately 37.0% and 31.5% of total respondents, respectively. No significant difference was observed in WTA among the different occupations. Conversely, gender, age, educational background, family size, and total arable land all had significant effects on WTA (Tables 3 and 4).

A multiple linear regression model was developed with WTA as the dependent variable, and gender ($X_1$), age ($X_2$), educational background ($X_3$), profession ($X_4$), family members ($X_5$), agricultural acreage ($X_6$), and land-area returned to forest ($X_7$), as the independent variables (Table 4). Results of the regression analysis suggest that the main factors affecting respondents' WTA are gender, age, and educational background (Table 5).

**Table 3.** Explanation of variables.

| Variable | Definition and Assignment |
|---|---|
| Gender | 1 = Male; 2 = Female |
| Age | 1 = < 30; 2 = 30–40; 3 = 40–50; 4 = 50–60; 5 = 60–70; 6 = > 70 |
| Profession | 1 = Farmer; 2 = Migrant Worker; 3 = Businessperson |
| Educational background | 1 = Illiterate; 2 = Primary School; 3 = Junior High School; 4 = High School; 5 = Undergraduate |
| Family size | 1 = 2–5; 2 = 6–8; 3 = > 8 |
| Agricultural acreage (ha) | 1 = 0–0.8; 2 = 0.8–1.5; 3 = > 1.5 |
| Farmland returned to forest (ha) | 1 = 0–0.3; 2 = 0.3–0.6; 3 = > 0.6 |

**Table 4.** Effects of social and economic variables on willingness to accept payment (WTA).

| WTA ($) | Gender | Age | Educational Background | Profession | Family Members | Agricultural Acreage (ha) | Farmland Returned to Forest (ha) |
|---|---|---|---|---|---|---|---|
| 0–221 | 1.000 ± 0.000 a | 3.390 ± 0.500 a | 1.220 ± 0.430 a | 1.000 ± 0.000 a | 1.000 ± 0.000 a | 2.000 ± 0.000 a | 2.000 ± 0.000 a |
| 222–444 | 1.150 ± 0.360 bd | 4.000 ± 0.000 b | 2.810 ± 1.080 b | 1.000 ± 0.000 a | 1.300 ± 0.470 b | 2.740 ± 0.450 b | 2.000 ± 0.000 a |
| 445–888 | 2.000 ± 0.000 c | 4.600 ± 0.550 c | 5.000 ± 0.000 c | 1.000 ± 0.000 a | 2.000 ± 0.000 c | 3.000 ± 0.000 b | 2.000 ± 0.000 a |
| 889–1331 | 2.000 ± 0.000 dc | 5.000 ± 0.000 dc | 5.000 ± 0.000 dc | 1.000 ± 0.000 a | 2.000 ± 0.000 dc | 3.000 ± 0.000 b | 2.570 ± 0.540 b |

Note: Different lowercase letters show differences across WTA groups.

**Table 5.** The main factors influencing farmer's WTA.

| Model | Standard Error | Standard Coefficient | T | Sig |
|---|---|---|---|---|
| Constant | 0.202 | _ | −9.484 | 0.000 |
| Gender ($X_1$) | 0.155 | 0.166 | 3.646 | 0.000 |
| Age ($X_2$) | 0.084 | 0.268 | 5.205 | 0.000 |
| Educational Background ($X_3$) | 0.098 | 0.226 | 6.652 | 0.000 |

## 4. Discussion

In China and elsewhere, stakeholders' WTA plays an important role in the implementation of afforestation policies, as is the case with other policies aiming to restore ecosystem services [57]. Family status, the income source of a farmer, state compensation efforts, and policy implementation have been found to have a significant influence on an individual's WTA [58]. In the current study, as shown in Table 4, no significant difference in WTA was observed between different occupations, in contrast to the findings of Zhang et al. [50]. The reason for this may be that the participants were all familiar with the income earned from farmland before its transformation to forests; their WTA for forestland was similar to this known cultivated land income. Conversely, gender, age, educational background, family size, and total arable land all significantly impacted participants' WTA (Tables 4 and 5).

Gender can affect people's behavior, values, and characteristics, and so can be an important factor in WTA [59]. The current study found that 24.0% of males and 40.2% of females had a WTA value higher than $444/ha/year (Table 1). This significant difference between genders is consistent with the research of Feng et al. [26], and may be related to the wage disparity between genders. As males are often on the upper end of income disparity, they are often less negatively impacted by income fluctuations. In the study region, females, however, tend to be engaged in the family's agricultural work [60]. The income from apricot forests (kernel collection), for example, would be mainly garnered from females' labor. As a result of this disparate impact, females in the study area may have been more prone to identify changes in income [61]. Age is another important factor affecting WTA. In the current study, the WTA value increased with respondents' age, contrary to the results of Feng et al. [26]. In the study area, it was observed that a significant portion of the young labor force in Huining County had moved from the region to find work elsewhere. These circumstances have created an environment in which the main labor force in the study area is over 50 years old; the survey demonstrated that 39.1% of the respondents were between 50 and 60 years old, and 33.7% were over 60 years old (Table 1). Based on the survey carried out in the present study, the primary purpose of land management for members of this generation is to ensure food security. Higher education was also found to be correlated with a higher WTA. These results are consistent with Li and Cai [62], but contrary to Feng et al. [26] and Wang [60]. A significantly negative relationship is often observed between educational background and WTA. This has been attributed to a more robust understanding of the long-term benefits of environmental protection among more educated participants [53]. The correlation between a higher level of education and a higher WTA in the current study suggests that at present, compensation in the region is not adequate to maintain the basic needs of participants and make up for the opportunity costs of afforestation, leading to a decline in living standards [63].

From an economic perspective, WTA is often affected by family income. For example, Yu and Cai [64] found that WTA decreased with increasing household annual income; this same trend is evident in the present study. The average WTA value for the Huining County respondents, at $777/ha/year, is significantly higher than that of respondents in other areas [58]. In the study area, participants' income was, on average, low ($195/year), and households seldom participated in markets outside of the agricultural sector. With high dependence on agriculture, changes in land use are especially impactful as they directly affect the primary income source for many residents. It is particularly important in such cases to instate an adequate compensation standard to offset the opportunity costs of afforestation. Some participants reported earning at least $666/ha/year from cultivated, pre-afforestation land, while the compensation from the national policy was $200–355/ha/year and the income from afforestation

was low (with most of the apricot kernels being used for food) - an income reduction, in extreme cases, of approximately 30%. However, previous studies have shown respondents' income after afforestation to vary regionally [65]. For example, in areas with poor agricultural productivity, subsidies were found to increase farmers' incomes, while in areas with higher yields, the same subsidies were insufficient to compensate for the opportunity cost of afforestation [66]. In terms of family size, the current study found that the larger the family, the higher the WTA, consistent with the research of Deressa et al. [65]. It is suggested that, when household income is derived principally from agriculture, the financial burden from losing income due to afforestation of previously cultivated land is exacerbated by additional family members [60,67]. Furthermore, it is likely that farmers with more arable land earn a higher revenue from farming, and thus report higher WTA values [53].

Besides the factors influencing WTA mentioned above, additional factors contribute to a high WTA in the study area. For example, populations of wildlife, such as hare, pheasant, and sparrow, have increased as the forest coverage in the study area has increased [68]. However, these improvements in biodiversity have initiated a series of adverse effects. All of the respondents reported that the number of wild animals feeding on their crops has increased rapidly since afforestation, resulting in damages to the crops and, in extreme cases, crop failure. In such cases, WTA increases to compensate for losses incurred as a result of afforestation.

## 5. Conclusions and Implications for Policy

In general, the RFFP has a substantial incubation period, and thus the return on investment is delayed. For example, apricot plantations are economic forests with a compensation period of five years. As a result, participants' income is often reduced in the short term due to the adjustment period necessary to establish an income-generating forest [69]. With the gradual growth of an economic forest, the economic benefits also emerge, and with proper management, the economic benefits may be higher after afforestation than when the same land was cultivated as traditional cropland [70]. However, in Huining County, the economic forests implemented in conjunction with the RFFP have seen a low survival rate. This, in combination with a lack of governmental management, has contributed to a low return on investment from the forest fruit industry, resulting in a gap between the income garnered from the cultivated land before afforestation, and the compensation and income earned from the economic forests after afforestation [71]. This situation contributed to the higher WTA observed in the study region. In this study, besides income, other social characteristics, such as gender, age, and educational level also had a significant influence on a farmers' WTA. In addition, the results of the current study suggest that the "one-size-fits-all" compensation standard is inadequate as it ignores the spatial heterogeneity of natural and economic conditions, resulting in a difference in satisfaction with the level of compensation, as found in other regions [72,73]. Therefore, in the study area, governments should pay locals at least an additional $422/ha/year to offset the gap between the average local WTA and the actual compensation given, by providing money through the rural credit cooperative, and recording payments in a passbook to enable farmers to verify how much they receive [31].

With this gap being unfilled, the risk of farmland reclamation remains largely unaddressed. This risk, it is argued, could be partially mitigated through comprehensive environmental policies [74]. For example, it may be possible to increase farmer income through the development of ecotourism, which could help alleviate compensation pressure [75]. Ecotourism activities, such as sightseeing among the apricot forests, picking gardens, youth agricultural plantations (planting organic wild vegetables), and rabbit hunting, could not only provide conditions for urban residents to experience rural life, but could also increase the income of farmers in the area and support the appropriate adjustment of the regional economic structure [76]. Farmhouse construction could also be leveraged to protect the environment by creating a new income source, reducing the pressure to cultivate land. This shift may allow some arable land to remain idle, leading to a decline in water consumption, and helping to improve the balance between the economy and ecology of the region [77].

In sum, to maintain the long-term benefits arising from afforestation, compensation should be specific-site dependent, and development measures based on local resources should be adopted by governments. Only with such changes can both ecological restoration and economic development be achieved.

**Author Contributions:** Conceptualization, J.C., X.W and J.F.A.; Methodology, J.C.; Validation, J.C.; Formal Analysis, X.W.; Investigation, X.W., G.W., X.D., G.Z. and C.L.; Data Curation, X.W. and J.Z.; Writing-Original Draft Preparation, X.W.; Writing-Review and Editing, J.C. and J.F.A.; Visualization, X.W.; Funding Acquisition, J.C.

**Funding:** This research was funded by the National Natural Science Foundation of China (41461109), the Major Program of the Natural Science Foundation of Gansu province, China (18JR4RA002), and the Key Laboratory of Ecohydrology of Inland River Basin, Chinese Academy of Science (KLERB-ZS-16-01) and the Open Fund for Key Laboratory of Land Surface Process and Climate Change in the Cold and Arid Region of the Chinese Academy of Sciences (LPCC2018008).

**Conflicts of Interest:** The authors declare no conflict of interest.

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
