# Peer review of "Farmers’ Willingness to Accept Compensation to Maintain the Benefits of Urban Forests"

_forests, doi:10.3390/f10080691_

Round 1

Reviewer 1 Report

Please make the abstract more focused on the research problem.

I suggest authors to review the introduction to make it more precise, especially the part of literature analysis and presentation of the aims of the study (paragraph 48-77).

Please clarify the method selected and why semi-structured questionnaire was done. Why 92 households were questioned. It is not clear why the authors have chosen the multivariate linear regression model, there are other possibilities to count WTA as well, which are more motivated by the literature.

I recommend English editing and spell checking.

Author Response

Comment

Please make the abstract more focused on the research problem.

Response

Thanks for your suggestion. We hope our revision can meet your requirement by adding the following sentences at the beginning of the abstract. In addition, other edits were also made in this section. After revision, we believe that the abstract more focuses on the research problem now.

P1, L 18-24: The Returning Farmland to Forest Program (RFFP) was implemented in China in 1999 with the goal of supporting environmental restoration by returning significant areas of cultivated land to forest. While afforestation supports long-term ecosystem services like carbon sequestration and the reduction of soil and water loss, it also reduces the amount of available arable land, putting financial pressure on those who depend on it for their livelihoods. In an effort to balance both ecological and economic benefits, regional governments offer financial compensation to farmers to offset these pressures in the form of the dollar amount per hectare of reforested land.

Comment

I suggest authors to review the introduction to make it more precise, especially the part of literature analysis and presentation of the aims of the study (paragraph 48-77).

Response

According to the comments from you and the reviewer 2, and the logical relationship of writing, we recognized this section. We invite you to look at it again because many edits were made and we can not present all of them to you here.

The aims of this study were rewritten as the following:

P3, L 470-474. Due to obstacles in PES schemes and CVM containing mechanisms to account for deviations as above-mentioned, in the current study CVM was selected to 1) explore the WTA of farmers from Huining county participating in the afforestation program; 2) determine the amount of compensation needed to fill the gap between previously offered subsidies and the WTA values determined by the locals; and 3) find the main factors influencing farmers’ WTA.

Comment

Please clarify the method selected and why semi-structured questionnaire was done. Why 92 households were questioned. It is not clear why the authors have chosen the multivariate linear regression model, there are other possibilities to count WTA as well, which are more motivated by the literature.

Response

In revised version, the reasons for method selected included two parts, reason for selection of CVM, and the reason for selection for WTA. The former was explained as: Although the CVM has inevitable deviation as a method of simulating the market[32], it contains mechanisms to reduce the impact of most deviations, such as providing the respondents with detailed information, increasing the credibility and certainty of the content of the questions, and controlling the investigation time of each sample at 20 to 30 minutes [36-38] (P2, L 222-226), and the latter was explained as: Based on the perspective of welfare loss, however, in developing countries and underdeveloped regions, such as Huining, labeled as a national-level poor county, it is recommended that the WTA of the affected party be used as a reference for the compensation standard, rather than the WTP [45] (P2, L 466-469).

The reason for semi-structured questionnaire is that a semi-structured questionnaire comprising both open-ended and multiple-choice questions was designed to counteract the biases inherent in single data sources [48] (P4, L 615-617).

92 households were questioned were done because In accordance with social survey requirements [35], 100 households were randomly selected as respondents from these 900, accounting for about 10% of the total households.92 valid surveys remained after removing households that had moved or that did not wish to participate (P4, L 620-623).

As for the reason for multivariate linear regression model chosen, we presented it as: Multivariate linear regression was used to analyze the effects of socio-economic variables on respondent WTA [18,55], as this approach has been determined to be suitable for analyzing the correlation between dependent and independent variables in similar contexts[56] (P 4, L 637-640) 

As for WTA calculation, it could be explained: Several methods have been used to calculate WTA based on, for example, the individual utility function theory [14], median [51], and Maximum Likelihood Estimation [52]. In the current study, the WTA is expressed as a weighted-average, a simple but accurate approach [53,54]. (P 4, L 632-634).

Comment

I recommend English editing and spell checking.

Response

This was heavily improved by our co-author from Canada. 

Reviewer 2 Report

General comment:

The topic of the study is relevant and interesting. I also think that it matches with the scope of a Special Issue of Forests entitled “The Benefits of the Urban Forest under Global Change”. In this form, however, the manuscript includes serious weaknesses and shortcomings. This is the reason why I recommend rewriting the manuscript and submitting it as a new manuscript to this journal.

Major concerns and suggestions:

In a survey, a CVM study creates a hypothetical market which is a reason why the results may include different types of biases. In this case study, it may have been possible to calculate the difference between the revenues of the original agricultural production and the revenues in the present situation with afforested land using market based information. Possibly added with an estimated sum of money to social security for migrants in urban areas, this would be a more reliable way to estimate for compensation to the landowners. Please, justify why in this case CVM still is an appropriate method to analyze the compensation amounts.

Introduction: Although the references of the section are reliable, they mostly are from China or on Chinese research. As this journal is international, I suggest broadening the view of the section with references on international research.

With respect to the discipline to the study, it is related to the research of Payments for Ecosystem Services (PES). This fact has not mentioned in the text and should shortly be referred in the introduction.

As an elicitation format, the study uses WTA type of question. In the literature, there is a lot of discussion about the disparity between WTP and WTA results. The authors should shortly refer to this literature (in Section 2.2, for example) and to justify the use of WTA in this case.

In Section 2, it would be informative to readers to tell what kind of possibilities farmers have to earn money with replanted forests. The authors do tell about apricot and almond trees on p. 6 and 7 but it would be informative to do this earlier and more general way.

Section 2.3: How do you test and develop the questionnaire before the survey? Please tell more.

Section 2: If possible, give some indications of the representativeness of the survey sample. Figures 3 and 4 are not very informative. Give the same information in a table and add to this table representative numbers about the population of the study area or the region.

The structure of Sections 3-5 should be reorganized so that Section 4 gives only the results of the analysis, not conclusion or discussion. In addition, Section 3 should divide into two sub-sections: 3.1 Descriptive results and 3.2. Valuation results. (The authors are free to formulate the exact headlines of the sub-sections.) In the former sub-section, the authors should present basic statistics of important variables and in the latter the results produced by the models. Section 3 should not include concluding comments or suggestions but they should be given in Section 4 named as “Conclusions and Discussion” or “Conclusions and policy implications”, for example. In Section 4, the author should repeat shortly the most important results and discuss them related to previous research and possible policy implications.

p. 6: Tales 1 and 2 are not very clear related both to format or the way of presentation and the explanation of the results. Clarify, for example, the explanation of the results of the analysis of variance.

In the present form, Section 5 (Conclusions and Suggestions) is problematic. Although the discussion and suggestions may be valid with respect to the study area and interesting for a reader the discussion and suggestions are mostly not based on the results of the study. Thus, could the authors limit the discussion and policy implications to the results of the study?

The paper would benefit from additional editing by a native English speaker. In fact, for example, I don’t fully understand sentences on p. 3, line 65-68: “He et al. (2016) [19] also found household’s livestock manure waste…” and p. 3, l. 107-109.

Minor suggestions:

The body text uses different fonts. Please use the same one.

After a decimal point of decimal numbers, it is better to have a one digit only, i.e. instead of 31.52% use 31.5%, for example.

page 2, line 65: after [19] “Also” should be “also”.

p. 3, Figure 1: The maps are small and the texts on the maps are small and unclear.

On p. 6, l. 178, you refer to Table 5 but you only have 2 tables.

p. 5 and p. 6: Variable names should be in the equal form both in the text and in the tables.

p. 6, l. 195: Delete Equation 1 as you show the same info in Table 2.

Author Response

Comment

The topic of the study is relevant and interesting. I also think that it matches with the scope of a Special Issue of Forests entitled “The Benefits of the Urban Forest under Global Change”. In this form, however, the manuscript includes serious weaknesses and shortcomings. This is the reason why I recommend rewriting the manuscript and submitting it as a new manuscript to this journal.

Response

Thanks for your apprise and confirmation. In previous version, it surely included serious weaknesses and shortcomings. According to you and the comments from the reviewer 1, we almost rewrote it.

Comment

1、In a survey, a CVM study creates a hypothetical market which is a reason why the results may include different types of biases. In this case study, it may have been possible to calculate the difference between the revenues of the original agricultural production and the revenues in the present situation with afforested land using market based information. Possibly added with an estimated sum of money to social security for migrants in urban areas, this would be a more reliable way to estimate for compensation to the landowners. Please, justify why in this case CVM still is an appropriate method to analyze the compensation amounts.

Response

Surely, a CVM may include different types of biases, but it can be mitigated. This can be achieved by providing the respondents with detailed information, increasing the credibility and certainty of the content of the questions, and controlling the investigation time of each sample at 20 to 30 minutes [36-38]. (P2, L 223-226).

In the study area, 81% of the respondents indicated that the apricot trees have no income, and 20% who had income forests by apricot kernels collection also were very low (about $4/ha/year) and most of apricot kernels were consumed by family based on our survey. In this case, as the specific income from forests is not clear, it is impossible to calculate inaccurately  the difference between the revenues of the original agricultural production and the revenues in the present situation with afforested land. Therefore, in main text, this information is lack. However, if possible, we can survey it in future. 

Cultivated land is only a social security guarantee for immigrant farmers when they lost their job in cities. If we consider money to social security for migrants, the WTA will be higher and inaccurate, because their WTA will be based on the city standard but not rural standard.

The reason using CVM asked also by the reviewer1 is that although the CVM has inevitable deviation as a method of simulating the market[32], it contains mechanisms to reduce the impact of most deviations, such as providing the respondents with detailed information, increasing the credibility and certainty of the content of the questions, and controlling the investigation time of each sample at 20 to 30 minutes [36-38] (P2, L 222-226).

Comment

2、Introduction: Although the references of the section are reliable, they mostly are from China or on Chinese research. As this journal is international, I suggest broadening the view of the section with references on international research.

Response

Thanks for your suggestion. In revised version, we added a few international researches, such as the following, to broaden the view of section:

Nyongesa J M, Bett H K, Lagat J K, et al. Estimating farmers’ stated willingness to accept pay for ecosystem services: case of Lake Naivasha watershed Payment for Ecosystem Services scheme-Kenya[J]. Ecological Processes, 2016, 5(1):15.

Swallow, B M., M. F. Kallesoe, U. A. Iftikhar, M. van Noordwijk, C. Bracer, S. J. Scherr, K. V. Raju, S. V. Poats, A. Kumar Duraiappah, B. O. Ochieng, H. Mallee, and R. Rumley. Compensation and rewards for environmental services in the developing world: framing pantropical analysis and comparison. Ecology and Society, 2009, 14(2): 26. [online] URL: http://www.ecology and society. org/ vol 14/ iss2/art26/.

Milder J C, Scherr S J, Bracer C. Trends and Future Potential of Payment for Ecosystem Services to Alleviate Rural Poverty in Developing Countries[J]. Ecology & Society, 2010, 15(2):4.

Kosoy N, Esteve Corbera. Payments for ecosystem services as commodity fetishism[J]. Ecological Economics, 2010, 69(6):1228-1236.

Cameron T.A. Contingent Valuation. In: Palgrave Macmillan (eds) The New Palgrave Dictionary of Economics. Palgrave Macmillan, London, 2008

Wegner & Irene G. Payments for ecosystem services (pes): a flexible, participatory, and integrated approach for improved conservation and equity outcomes. Environment, Development and Sustainability, 2016, 18(3), 617-644.

Mahiea P H, Kristrom B, Brannlund R, Giergiczny. Exploring the determinants of uncertainty in contingent valuation surveys[J]. Journal of Environmental Economics and Policy, 2014, 3(2):186-200.

Douglas MacMillan, et a1. Costs and benefits of wild goose conservation in Scotland. Biological Conservation, 2004, 119: 475-485

Dargiri M N, Shamsabadi H A, Thim C K, et al. Value-at-risk and Conditional Value-at-risk Assessment and Accuracy Compliance in Dynamic of Malaysian Industries[J]. Journal of Applied Sciences, 2013, 13(7): 974-983.

Koetse M J. Effects of payment vehicle non-attendance in choice experiments on value estimates and the WTA-WTP disparity[J]. Journal of Environmental Economics and Policy, 2017, 6(3):225-245.

Flachaire E, Guillaume Hollard, Jason F. Shogren. On the origin of the WTA–WTP divergence in public good valuation[J]. Theory & Decision, 2013, 74(3): 431-437.

Comment

3、With respect to the discipline to the study, it is related to the research of Payments for Ecosystem Services (PES). This fact has not mentioned in the text and should shortly be referred in the introduction.

Response

Surely, it should be mentioned shortly in the text. In revised version, we added its definition and the limitations as following(P2, L 201-214):

PES rewards resource managers for the provision of ecosystem services through cash payments or other compensation based on market, and is characterized by ecological functions subject to trade, the standard of exchange, and supply and demand between seller and buyer of these services [22-24]. Although it was widely used to estimate compensation (e.g. [23]), Wegner and Irene [25] found two major obstacles in the design and implementation of effective PES schemes. The first is that PES fails to account for both the physical properties of ecosystem services and the factors influencing the capacity of land users and ecosystem service beneficiaries to participate in PES and derive benefits from it. According to them, this arises from the focus of PES on maximization of economic efficiency, and the assumption that individuals respond to financial incentives uniquely on the basis of personal utility calculations. The second obstacle is that PES schemes tend to be disjointed from broader national strategies for rural development and from wider socio-economic trends that strongly influence the capacity of land users to participate in these schemes and shift to sustainable land-use practices. In practice, both obstacles are difficult to overcome.

Comment

4、As an elicitation format, the study uses WTA type of question. In the literature, there is a lot of discussion about the disparity between WTP and WTA results. The authors should shortly refer to this literature (in Section 2.2, for example) and to justify the use of WTA in this case.

Response

Thanks for this good suggestion. In revised version, we added contents related to WTA and WTP as following (P2,3  L 227-469):

Willingness to pay (WTP) and willingness to accept pay (WTA) can be used, in both approaches, as proxy economic measures of environmental services, with the former being more relevant for buyers of ecosystem services (i.e. ‘acquisition’), and the latter being more relevant for sellers of ecosystem services (i.e. ‘loss’) [14,39-41]. Generally, WTP is lower than WTA [42]. Low WTP is primarily attributed to the limited economic value of ecosystem services provided by individual land owners [23], while higher WTA is due to high costs associated with significant changes in the land use, and participant identities and lifestyles [43,44]. Based on the perspective of welfare loss, however, in developing countries and underdeveloped regions, such as Huining, labeled as a national-level poor county, it is recommended that the WTA of the affected party be used as a reference for the compensation standard, rather than the WTP [45].

Common

5、In Section 2, it would be informative to readers to tell what kind of possibilities farmers have to earn money with replanted forests. The authors do tell about apricot and almond trees on p. 6 and 7 but it would be informative to do this earlier and more general way.

Response

Based on survey, in apricot forests, income is only from apricot kernels by collection. We added this information on P8, L 1315.

In previous manuscript, we did not clearly present the difference between apricot tree and apricot kernels. In fact, almond trees refer to apricot trees. In order to be clear, in revised version, we replaced ‘almond’ with ‘kernel’.

Comment

6、Section 2.3: How do you test and develop the questionnaire before the survey? Please tell more.

Response

In revised version, more details about testing and developing the questionnaire before the survey were introduced as following (P4, L 613-615):

From late July to early August of 2017, a total of 92 households were surveyed. First, 20 households participating in the RFFP and a few local staff members engaged in this project were interviewed with an open-ended questionnaire. Based on these collected data, a semi-structured questionnaire comprising both open-ended and multiple-choice questions was designed to counteract the biases inherent in single data sources [48]. Next, a pre-survey of 15 households was conducted to identify potential improvements in the developed questionnaire. Finally, after modification and finalization of the questionnaire, the formal investigation was carried out. Roughly 900 households were located in the study area (Fig 1).

Comment

7、Section 2: If possible, give some indications of the representativeness of the survey sample. Figures 3 and 4 are not very informative. Give the same information in a table and add to this table representative numbers about the population of the study area or the region.

Response

In order to present more information about survey samples, Figures 2-4 were changed to Table 1 and Table 2 (P4, 653-933; P5, 945-1100). In addition, the total households and population in the study area were presented as: roughly 900 households were located in the study area (Fig 1) (P3, L 488-489).

Comment

8、The structure of Sections 3-5 should be reorganized so that Section 4 gives only the results of the analysis, not conclusion or discussion. In addition, Section 3 should divide into two sub-sections: 3.1 Descriptive results and 3.2. Valuation results. (The authors are free to formulate the exact headlines of the sub-sections.) In the former sub-section, the authors should present basic statistics of important variables and in the latter the results produced by the models. Section 3 should not include concluding comments or suggestions but they should be given in Section 4 named as “Conclusions and Discussion” or “Conclusions and policy implications”, for example. In Section 4, the author should repeat shortly the most important results and discuss them related to previous research and possible policy implications.

Response

Thanks for your good suggestion. Section 3 was reorganized into three sub-sections in order to make it more clear and logical. If we put all things into section 4, it is too long, and unreadable. In this case, we listed a single section named as ‘Discussion’ (Section 4 in revised version), then section 5 is “Conclusions and policy implications” in revised version. We hope these changes can meet your requirements.

Comment

9、p. 6: Tales 1 and 2 are not very clear related both to format or the way of presentation and the explanation of the results. Clarify, for example, the explanation of the results of the analysis of variance.

Response

The previous Tables 1 and 2 are now Table 4 and Table 5 (P7,8 L 1276-1298). In revised version, in order to present more clearly variables, we added a new Table 3 (P6, 1114-1115). In addition, the explanation of the results of the analysis of variance was presented in section 3.3 (more details on P6, L 1101-1108).

Comment

10、In the present form, Section 5 (Conclusions and Suggestions) is problematic. Although the discussion and suggestions may be valid with respect to the study area and interesting for a reader the discussion and suggestions are mostly not based on the results of the study. Thus, could the authors limit the discussion and policy implications to the results of the study?

Response

Thanks for your suggestion. We have deleted sentences that were not related to the results of the study, and limited the discussion and policy implications to the results of the study. However, we retained some sentences that related to increasing income of locals and decreasing financial burden of the governments, in order to deepen and widen the implications of our study. This section was almost written, so please look at it on P9, L1557-1824.  

Comment

11、The paper would benefit from additional editing by a native English speaker. In fact, for example, I don’t fully understand sentences on p. 3, line 65-68: “He et al. (2016) [19] also found household’s livestock manure waste…” and p. 3, l. 107-109.

Response

Surely, one of our co-author who is a native English speaker edited carefully again.

Comment

Minor suggestions:

The body text uses different fonts. Please use the same one.

After a decimal point of decimal numbers, it is better to have a one digit only, i.e. instead of 31.52% use 31.5%, for example.

page 2, line 65: after [19] “Also” should be “also”.

p. 3, Figure 1: The maps are small and the texts on the maps are small and unclear.

On p. 6, l. 178, you refer to Table 5 but you only have 2 tables.

p. 5 and p. 6: Variable names should be in the equal form both in the text and in the tables.

p. 6, l. 195: Delete Equation 1 as you show the same info in Table 2.

Response

All of these were corrected or redone. 

Round 2

Reviewer 1 Report

Thank you for improvements of the article, especially for clarification of the method selected.